# Periodontal Healing with Fixed Restorations Using the Biologically Oriented Preparation Technique Combined with a Full Digital Workflow: A Clinical Case Report

**DOI:** 10.3390/healthcare11081144

**Published:** 2023-04-16

**Authors:** Tommaso Rinaldi, Andrea Santamaría-Laorden, Jaime Orejas Pérez, Laura Godoy Ruíz, Carlos Serrano Granger, Pablo Gómez Cogolludo

**Affiliations:** Department of Clinical Dentistry, Faculty of Biomedical and Health Sciences, European University of Madrid, 28670 Madrid, Spainpablo.gomez@universidadeuropea.es (P.G.C.)

**Keywords:** BOPT, fixed restorations, intraoral scanners, optical impression, periodontal healing, supracrestal tissue attachment

## Abstract

(1) Gingival inflammation is an ongoing challenge in tooth-supported fixed restorations, especially when the prosthetic margin does not consider the supracrestal tissues of patients. This case report aimed to present the case of a patient who was periodontally compromised due to a previous invasion of the supracrestal tissue attachment with fixed restorations and evaluate the healing response of periodontal tissues to a vertical edgeless preparation technique: bleeding upon probing (BOP), periodontal probing depth (PPD) and clinical attachment level (CAL). (2) After tooth preparation, the new restorations were adapted, this time without invading the supracrestal space of the patient, and CAD/CAM monolithic zirconia crowns were fabricated. (3) Optimal maturation of the soft tissue was observed, achieving correction of the marginal contour of periodontal tissues and improvement of periodontal indexes. (4) It can be concluded that the BOPT technique combined with a full digital workflow is a valid option for the correction and remodeling of gingival architecture.

## 1. Introduction

Traditionally, prosthetic preparations have always had a finishing line, also called horizontal preparation—be it a shoulder, shoulder with bevel, or chamfer. However, the apical migration of the gingival margin remains a recurrent complication for this type of preparation, resulting from inadequate quality and quantity of keratinized gingiva, reaction to trauma during prosthetic work, or chronic inflammation due to prosthetic errors [1]. Hence, Loi and Di Felice introduced the biologically oriented preparation technique (BOPT) with the aim of guiding the soft tissue through a prosthetic restoration with a vertical preparation [2].

This periodontal response is enabled by the correct management of the prosthetic margins of our restorations, which help control the wound and bleeding induced by a subgingival tooth preparation. Therefore, the healing of the socket will adapt to the new restoration, differing from the conventional technique, in which the crown had to adapt to the periodontal architecture of the patient [3].

However, it is important to consider the anatomy of the dentogingival complex before approaching any case with such a technique. As described by Ahmad, when looking at a healthy periodontium, we can clearly distinguish three structures or layers which make up the supracrestal tissue attachment of the gingiva: the supracrestal connective tissue attachment, measuring 1.07 mm; epithelial (or junctional epithelium) attachment, which measures 0.97 mm; and a 0.67 mm sulcus [4,5].

The most important factor to preserve the dimensions of our periodontal tissues and avoid the apical migration of the gingival margin is the concept of supracrestal tissue attachment. Traditionally called biological width, it is described as the sum of the connective tissue and epithelial attachment, which roughly measures 2.04 mm [6,7]. It is a natural barrier that protects the two most vulnerable structures of a tooth, the periodontal ligament and the alveolar bone, and is delimited by the cemento–enamel junction (CEJ) [4].

Complications resulting from the invasion of the supracrestal tissue attachment may include chronic inflammatory gingival response around the restoration, bleeding upon probing, localized gingival hyperplasia with minimal bone loss, gingival recession, deep sockets formation, clinical attachment loss and loss of alveolar bone [8].

Traditionally, the treatment in patients with fixed restorations with iatrogenic invasion of supracrestal tissue attachment would imply a resective periodontal surgery with crown lengthening and gingivectomy, restoring the proper distance between the horizontal preparation and the alveolar ridge.

The BOPT technique, by means of a vertical preparation, makes it possible to reposition the margin of our restoration with a more conservative approach, which allows for guided and controlled periodontal healing of the soft tissues. Rodriguez and co-workers (2019) describe this process as a reorganization of the connective and epithelial tissue in the cement of our tooth, limited by the design of a restoration [9].

### Description of the BOPT Technique

The biologically oriented preparation technique (BOPT) described by Loi is an extremely sensitive technique and must follow precise steps. Before starting the procedure, probing is performed to determine the level of existing epithelial attachment. During the initial phase of the procedure, the coronal part of the tooth must be prepared, performing an intrasulcular preparation with diamond fine burs (100–120 µm) to cause an intrasulcular wound (Figure 1) [2].

Thus, the cementoenamel junction is eliminated. Temporary acrylic crowns, previously prepared by the technician, are relined with acrylic resin (Figure 2), with the aim of creating a new contour for the crown and stabilizing the clot, and they are positioned in the sulcus, with a depth of 0.5–1 mm from the gingival margin (Figure 3), respecting the biologic width of the soft tissue (controlled invasion of the gingival sulcus) [2].

At this point, the presence of a new periodontal ligament with connective tissue can be observed, with connective tissue immersed in the newly formed cement in the prepared tooth area [9].

After the maturation of the soft tissue is complete (a minimum of 4 weeks), it will be possible to take impressions to complete the restoration. The technician will then place the crown margin 1 mm below the gingival margin, ensuring that the crown will never invade the junctional epithelium (Figure 4) [2].

## 2. Materials and Methods

We present the case of a 43-year-old non-smoking patient with no significant medical history, whose recent rehabilitation (3 months old) of the aesthetic front (teeth 1.3-1.2-1.1-2.1-2.2-2.3) concurred with abundant spontaneous bleeding and a severe case of localized stage III grade C periodontitis with gingival hyperplasia caused by the invasion of the restoration margin into the supracrestal space in the aesthetic region (Figure 5a). Upon probing, periodontal indexes were determined from canine to canine in the maxillary arch, with an average probing pocket depth (PPD) of 4.5 mm, probing attachment levels (PAL) of 4.58 mm, plaque index (PI) of 32% and bleeding on probing (BOP) of 26%. We observed localized deep pseudopockets up to 7 mm on the lateral and central incisors (Figure 5b).

A series of periapical radiographs were performed (Figure 5c) to rule out a case of generalized chronic periodontitis. After anamnesis and intraoral examination, we determined that the old restorations had violated the supracrestal tissue attachment, triggering the acute inflammatory response.

A new treatment plan was established, prepping teeth with an edgeless vertical margin with the biologically oriented preparation technique and placing new monolithic zirconia crowns.

### Clinical Sequence

A 3Shape TRIOS 3 intraoral scanner (3Shape^®^, Copenhagen, Denmark) was used to perform an intraoral scan prior to preparation of new crowns (Figure 6a). It was demonstrated by Orejas-Pérez and colleagues (2022) that TRIOS offers satisfactory precision as an optical impression system. The scanning was sent to the technician for the preparation of PMMA milled temporary crowns [10,11].

After removing the old restorations, the presence of cement was observed in the patient’s sulcus, which further explains the inflammatory response of the periodontal tissues (Figure 6b).

Therefore, trimming and removal of the excess cement at the intrasulcular level was performed with a fine periosteal diamond bur. The teeth were then prepared using diamond burs of 100–120 µm (Figure 6c).

During the preparation, bone-level gingitage was performed as described by Rodriguez and co-workers (2019) with 1.2 mm fine diamond burs, causing intentional bleeding controlled by the provisional crowns sealing the intrasulcular wound at the marginal level of the gingiva (Figure 6d) [9].

The cemento–enamel junction of the natural tooth, which, by this point, was eliminated, was then transferred to a cemento–crown line, since the crown is responsible for modifying and controlling the healing process of the tissues.

At this point, the laboratory-prepared polymethyl methacrylate acrylic (PMMA) milled temporary crowns with a contour that followed the gingival margin based on the previously performed scan. In the present clinical case, the decision was made to fabricate the provisional crowns following the biological concept preserving the supracrestal space in order to stabilize the margins 3 mm away from the alveolar ridge (Figure 7a).

The provisional crowns were adapted using a transparent vacuum shell. Autopolymerizing acrylic resin was used to fill the small gap between the tooth and the fitting surface of the provisional restoration. Before full polymerization, excess buccal, interproximal and palatal resin were removed with a thin probe. After full polymerization, the provisional restoration was removed from the mouth, the internal finish line was marked with a pencil and the remaining space was filled with acrylic or composite resin. The provisional restoration was finished and polished.

The recommended waiting time after this procedure is 1 month, which is the minimum period needed for tissue maturation. However, the provisional crowns were left untouched for 2 months to let the inflammatory and hypertrophic condition subside.

At the review appointment, we appreciated how the intrasulcular portion of the temporary crown margin stabilized the fully structured gingival tissue (Figure 7b). Regular probing can now be performed without bleeding, and a new large junctional epithelium has formed. Creeping attachment can be observed clinically when lifting the provisional crowns on tooth 1.1, which is the concentration of blood vessels below the provisional—proof that there is neo-formation of healthy periodontal tissues in the cervical area (Figure 8).

A final intraoral scan was performed, allowing the technician to design the full-coverage monolithic zirconia crowns with exocad CAD software 3.0 (Figure 9). Thereafter, a plastic trial was performed for a final aesthetic analysis of the shape of the crowns and the gingival margins (Figure 10a).

The plastic try-in was then scanned to transfer the design to the final restoration. Thereafter, the definitive crowns were milled from a monolithic zirconia A2 shade block (Prettau; Zirkonzahn; Gais, South Tyrol, Italy).

A final appointment for crown delivery and cementation was held. The crowns were cemented with glass ionomer cement; as shown below, complete healing of the soft tissues can be observed (Figure 10b).

## 3. Results

The healing proceeded uneventfully. Clinically, 8 weeks after the new preparation and temporization, soft tissues presented a good healing and optimal volume and contour of the gingival margin. Furthermore, ongoing maturation can be observed in the papilla area between the preparation, with a soft tissue gain. Periodontal indexes also improved, with an average PPD of 3.22 mm, PAL of 3.28 mm, PI of 8% and BOP of 11%. This was clearly linked to the conditioning of the temporary crowns. The volumetric gain obtained with the technique appears clinically stable with a healthy texture of the restored papilla.

## 4. Discussion

Periodontal complications associated with prosthetic treatments can result from multiple etiological factors, such as mechanical forces, inflammatory response, iatrogenic factors, anatomical malposition, inadequate quality and quantity of keratinized gingiva, chronic inflammation due to prosthetic errors, etc. [4,12]. Eliminating the etiological factors will be key to the coronal migration of the gingival margin [1].

Traditionally, a more invasive approach would have been suggested to solve a case of supracrestal tissue attachment invasion with fixed tooth supported prostheses such as the one presented. However, the BOPT technique has been proven to be able to produce a periodontal regeneration on the dentin surface of the previously carved tooth [13]. Therefore, BOPT should not only be considered a prosthetic treatment, but also and foremost a regenerative periodontal treatment, due to the sulcus de-epithelialization and the space-maintaining and clot-preserving properties of the prosthesis [9].

Hence, the BOPT technique allows the gingiva to migrate coronally, solving complications such as apical migration and recession of the soft tissues that occur in the vast majority of cases, as described by Vehkalahti (1989) [14].

To understand the biologic response shown in the present case report, it is important to understand tissue healing. The healing process is based on the following phases: hemostasis, followed by inflammation, cell proliferation and remodeling [15,16]. These phases concur in the biologically oriented preparation technique, remodeling the soft tissue [3,16].

In the case of healing by secondary intention, such as the one treated, part of the fibroblasts constitutes the connective tissue change into myofibroblasts [9]. This will bring the gingival margins closer, and therefore all the tissue and fibers that surround the prepared tooth will contract in the direction of the smaller diameter—in this case, coronally [3,16]. After one month, the myofibroblasts will disappear, and the denuded root can stimulate the differentiation of cementoblasts, which will lay down strong tissue to which the new collagen fibers can attach. This highlights the importance of the provisional in this technique, since it will be the one in charge of stabilizing the clot so that the healing process can develop and stabilize [15,16].

As shown by Rodriguez and co-workers (2019), the vertical preparation in BOPT leads to a better coronal seal and contour [9]. This allows for less cement leakage and less bacterial filtering, which demonstrates the importance of placing the margin of the crown within the sulcus [2]. This will help adapt the anatomy and architecture of the gingiva around the forms the clinician has planned.

Traditional restoration with horizontal finish lines is associated with apical migration of gingival margin and periodontal complications [17,18]. This study presents an optimal healing outcome and correction of the gingival contour, with no spontaneous bleeding on probing and no clinical sign of inflammation. Pelaez and colleagues (2012), in a four-year prospective study, showed a negative response of the gingival margin, where 89.47% of the subjects suffered apical migration of the gingival margin in preparations with a subgingival horizontal finish line [19].

As for the material used for the restoration, Agustín-Panadero and co-workers (2016) performed a prospective clinical study with zirconia full-coverage restorations on teeth prepared with BOPT, in which a 100% stability of the gingival margins was obtained, with only 2% registered complications rate of the treated teeth and an overall survival rate of 100% with a 2-year follow-up [20]. The combination of zirconia with a full digital workflow allowed for a conservative approach, as we were able to reduce the tissue manipulation with a more invasive approach with traditional techniques and conventional materials such as polyethers or silicone-based elastomers [10].

As a limitation of the present study, however, the absence of long-term clinical studies on the technique makes it difficult—if not impossible—to compare the results obtained in this study with consistent existing literature.

## 5. Conclusions

The BOPT technique, combined with a full digital workflow, can be considered a valid alternative treatment and predictable solution for soft tissue correction and remodeling in patients with periodontal defects and inflammation in the aesthetic area. The re-establishment of an ideal and healthy contour of the gingival margin provided both aesthetic and functional improvements, maximizing tissue maturation to prevent gingival recessions over time.

## Figures and Tables

**Figure 1 healthcare-11-01144-f001:**
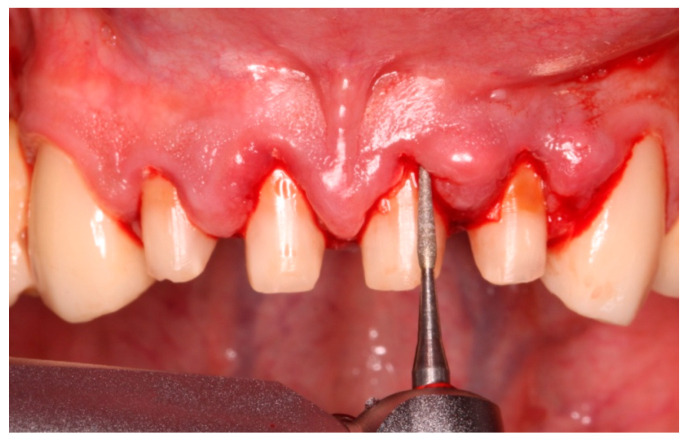
Intrasulcular vertical preparation with diamond burs of 100–120 µm. The cemento–enamel junction is completely removed, and bleeding can be observed.

**Figure 2 healthcare-11-01144-f002:**
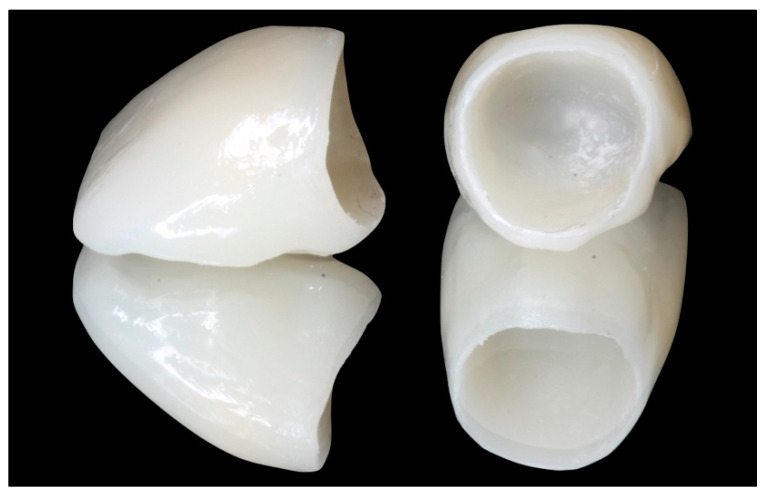
Prosthetic emergence profile of provisional restorations that will seal the clot and favor periodontal healing.

**Figure 3 healthcare-11-01144-f003:**
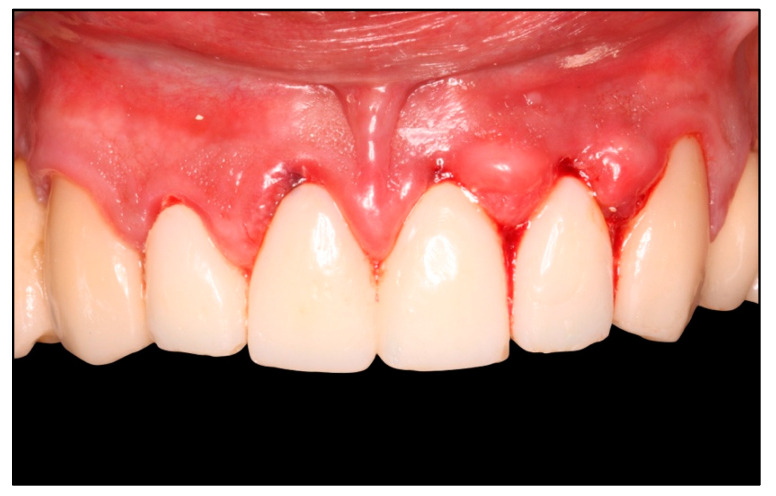
Adaptation of the provisional crowns 1 mm below the gingival margin to prevent invasion into the supracrestal space.

**Figure 4 healthcare-11-01144-f004:**
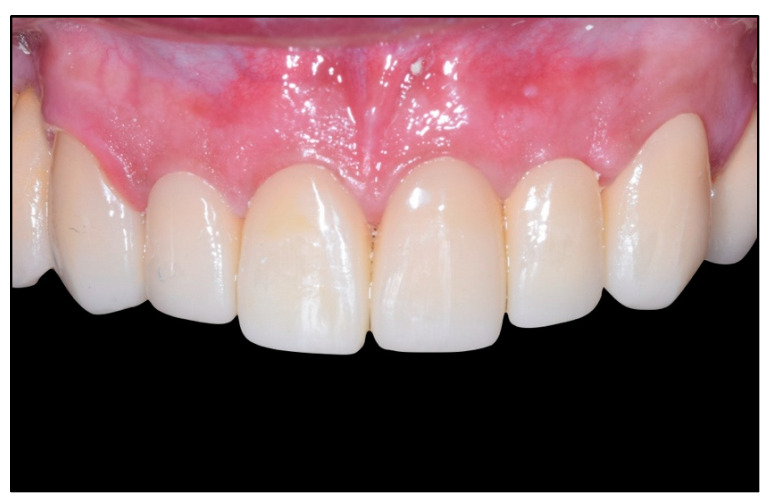
Final restorations. After periodontal maturation, the gingival architecture is restored, and full maturation of soft tissues can be observed.

**Figure 5 healthcare-11-01144-f005:**
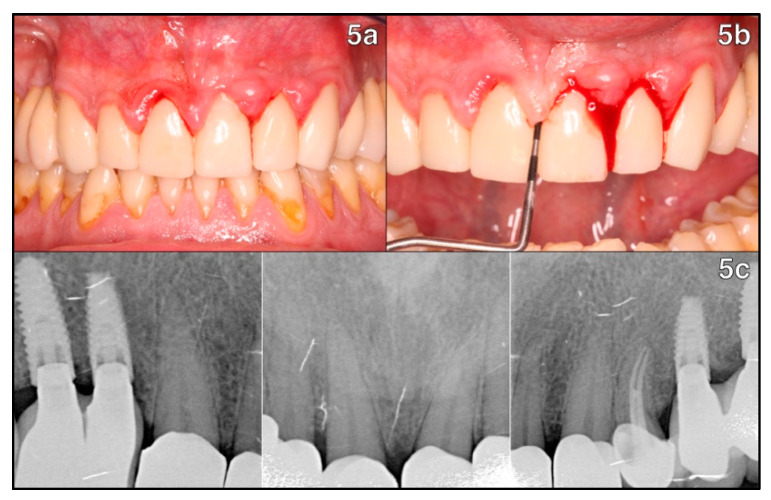
Intraoral examination of teeth 1.3-1.2-1.1-2.1-2.2-2.3: (**5a**) pockets and spontaneous bleeding are evident; (**5b**) profuse bleeding on probing can be observed, as well as deep pseudosockets up to 7 mm on teeth 12-11-21-22; (**5c**) periapical radiographs in the aesthetic region which show implant supported prostheses on teeth 1.5-1.4-2.4-2.5, root canal treatment in 2.3 and full coverage crowns on 1.3-1.2-1.1-2.1-2.2-2.3, with localized bone resorption between 1.1-2.1 and 2.1-2.2.

**Figure 6 healthcare-11-01144-f006:**
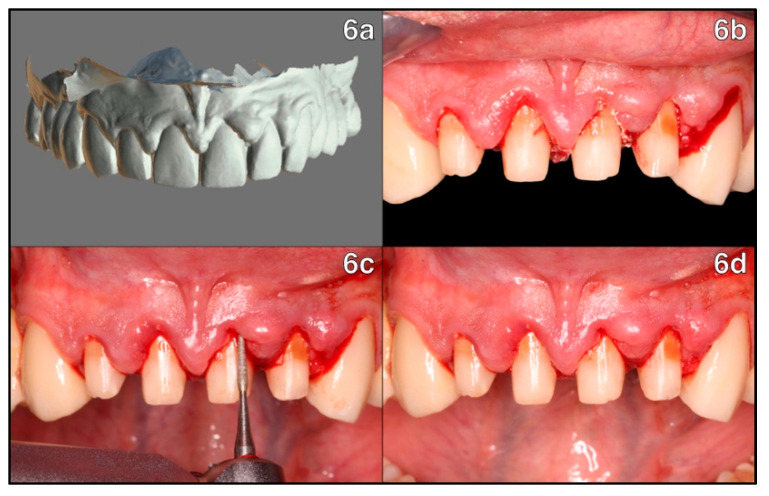
Intraoral scan pre-op and removal of old restorations: (**6a**) 3D reproduction of the arch after intraoral scanning, performed before old restorations are removed; (**6b**) after removal of the old restorations, intrasulcular cement can be observed in the sockets of teeth 1.1-2.1-2.2; (**6c**) trimming and removal of excess cement with fine periosteal diamond bur; (**6d**) frontal view of teeth after preparation.

**Figure 7 healthcare-11-01144-f007:**
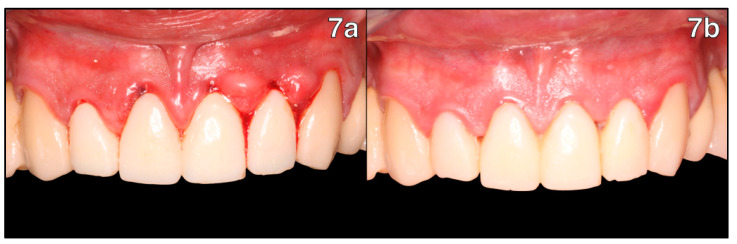
Temporization and periodontal tissue healing of new preparations: (**7a**) adaptation of new temporary crowns; (**7b**) review appointment 2 months after temporary crowns placement; optimal periodontal healing can be observed.

**Figure 8 healthcare-11-01144-f008:**
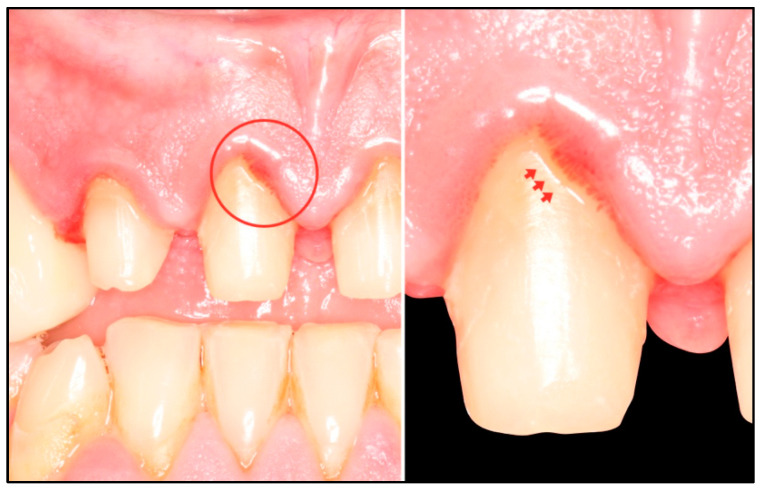
After lifting the temporary crowns, creeping attachment can be observed at the marginal level of the soft tissues.

**Figure 9 healthcare-11-01144-f009:**
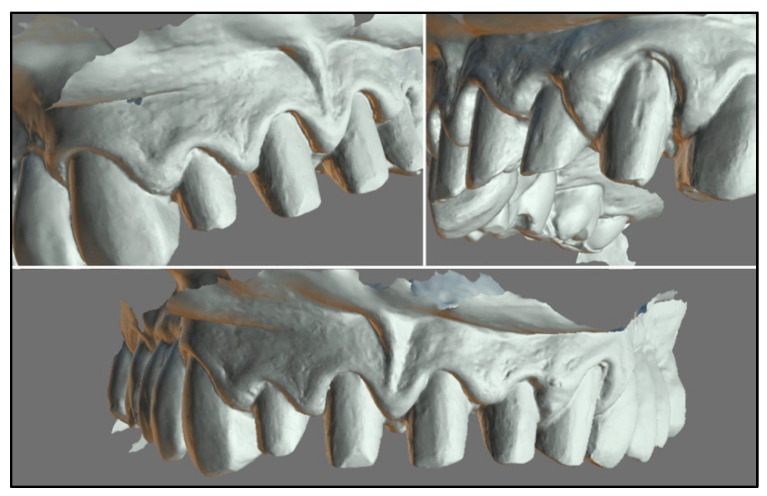
Intraoral scanning post-healing. Optimal definition can be observed at the marginal level of each preparation.

**Figure 10 healthcare-11-01144-f010:**
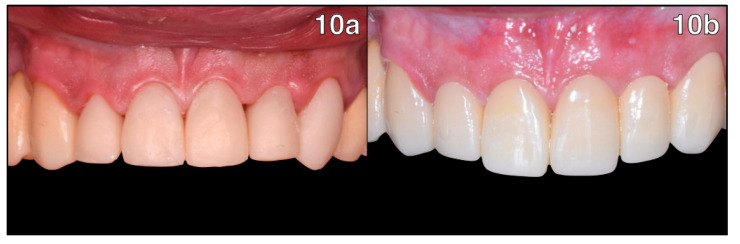
Plastic try-in and delivery: (**10a**) ideal contouring of the gingival architecture and healthy periodontal tissues can be observed; (**10b**) final delivery of the full-coverage monolithic zirconia restorations.

## Data Availability

The data presented in this study are available upon request from the corresponding author. The data are not publicly available due to privacy.

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
