# Peer review of "Periodontal Healing with Fixed Restorations Using the Biologically Oriented Preparation Technique Combined with a Full Digital Workflow: A Clinical Case Report"

_healthcare, 2023, doi:10.3390/healthcare11081144_

Round 1
Reviewer 1 Report
The topic of the article is interesting from a clinical standpoint.
There are some comments below:
- The case presentation don’t have sufficient date. It is necessary to add the prosthodontic diagnostic, occlusion diagnostic. It is not specified if the patient has a good hygiene, is healthy or unhealthy. We do not know how many teeth are evaluated.
- Which tooth has 7mm deep pocket?
- How were performed the provisional crowns? It is not clear. Also, the zirconia crowns – please, describe the entire steps for performing them.
Author Response
Thank you very much for your comments. We added the requested data, periodontal indexes have been described, medical history and hygiene of the patient as well. Indeed, we realized some concepts were not clear enough. You may find the changes applied in the PDF attached at lines: 97 to 113; 150 to 156; 179 to 181.

Reviewer 2 Report
This case report has its merits trying to highlight the application of digital workflow in BOPT in a patient with localised periodontitis. Kindly refer to the Comments annotated in the pdf file. In addition, there are consistent grammatical mistakes throughout the manuscript that require attention.

Author Response
Thank you very much for your comments, they have been extremely helpful. We have corrected and sent the paper to a translator for grammar check and reviewing. Please see the PDF below where we highlighted all corrections requested. We also changed, corrected and added more recent references.
We would like to apologize in advance for the following:
- Regarding the rearrangement of the introduction, we could not mention and explain thoroughly the BOPT technique earlier in the introduction as you asked, since one of the main concerns of another reviewer was precisely that we needed to give the BOPT technique explanation more space, including pictures of a typical case in the paragraph 3.1 (which we have corrected naming it “1.1”), we are truly sorry we could not include this correction in our paper as we did with all the others corrections you asked.
- Furthermore, thanks to your advices and after applying all changes and corrections, we now see the title better reflects and suits the text, we truly appreciate your comments.
Every other suggestion and grammar correction was included. Thank you again for your insights.

Reviewer 3 Report
This paper is a digital application of the BOPT technique to a patient with periodontitis, but I consider it unacceptable because there is no reason to apply the full digital technique and it is difficult to understand the usefulness of the BOPT technique itself from this case report.
1.Introduction
3.1 is incorrect. Please correct it.
Regarding the effectiveness and sensitivity of the BOPT technique, please describe not only the significance and purpose of the BOPT technique with references and brief descriptions, but also describe the treatment steps with figures for a typical case.
2. Results
  The results section is very poor with only pictures and comments. Gingival inflammation, PPD, esthetic score, etc. need to be described.
 Also, please explain in the Discussion why the BOPT technique is the reason why the results are good including esthetics.

3.Discussion
The reason why the BOPT technique was effective in this case and the reason why the full digital workflow was used and its effectiveness are not clearly stated.
In the Discussion section, please clearly describe the effectiveness of the BOPT technique used in this case, not in general terms.
Author Response
- Thank you very much for your comments. We have corrected and included as suggested the steps with figures of the case in paragraph “1.1 Description of the BOPT technique” (see PDF attached).
- Again, thank you very much for your comments. We have added periodontal indexed and explained why BOPT technique is the reason why the results are optimal on lines 97 to 113 and 207 to 216 (you can find all changes in the PDF attached).
- Thank you very much for all your insights. We would like to point out that the digital workflow is just an instrument, and as such it was complementary in achieving the results obtained in our case report. By using extremely compatible materials such as monolithic zirconia and a conservative approach with intraoral scanners, we are able to reduce stress and manipulation of the soft tissues as well as reducing a more invasive approach with conventional impression materials. We added the following lines to further explain our point: 243 to 250 (again you may find all changes in the PDF attached).

Round 2
Reviewer 2 Report
Please see the comments and suggestions indicated in the pdf file.

Author Response
We appreciate your comments. We have sent our document for extensive English revision and editing (you can find the attached certificate below) and have included the suggested modifications. Thank you very much for your insights.

Reviewer 3 Report
The article provides a detailed explanation of the BOPT technique and the significance of the digital workflow, including references and figures, and is considered to be an acceptable case report.
Author Response
We appreciate your insight and are happy you enjoyed our case report. Thank you very much for your insight. We have added further modifications as suggested by reviewers. Regards.
